# Burning Behaviour of Rigid Polyurethane Foams with Histidine and Modified Graphene Oxide

**DOI:** 10.3390/ma14051184

**Published:** 2021-03-03

**Authors:** Kamila Sałasińska, Milena Leszczyńska, Maciej Celiński, Paweł Kozikowski, Krystian Kowiorski, Ludwika Lipińska

**Affiliations:** 1Department of Chemical, Biological and Aerosol Hazards, Central Institute for Labour Protection—National Research Institute, 00-701 Warsaw, Poland; maciej.celinski@ciop.pl (M.C.); pawel.kozikowski@ciop.pl (P.K.); 2Faculty of Materials Science and Engineering, Warsaw University of Technology, 02-507 Warsaw, Poland; milena.leszczynska.dokt@pw.edu.pl; 3Department of Chemical Synthesis and Flake Graphene, Łukasiewicz Research Network—Institute of Microelectronics and Photonics, 01-919 Warsaw, Poland; krystian.kowiorski@imif.lukasiewicz.gov.pl (K.K.); Ludwika.Lipinska@imif.lukasiewicz.gov.pl (L.L.)

**Keywords:** polyurethane foam, graphene oxide, burning behavior, fire retardant

## Abstract

Since rigid polyurethane (PU) foams are one of the most effective thermal insulation materials with widespread application, it is an urgent requirement to improve its fire retardancy and reduce the smoke emission. The current work assessed the fire behavior of PU foam with non-halogen fire retardants system, containing histidine (H) and modified graphene oxide (GO_A_). For investigated system, three loadings (10, 20, and 30 wt.%) were used. The Fourier transform infrared spectroscopy (FT-IR), differential scanning calorimetry (DSC), thermogravimetric analysis, cone calorimetry (CC) and smoke density chamber tests as well as pre- and post-burning morphological evaluation using scanning electron microscope (SEM) were performed. Moreover, TGA combined with FT-IR was conducted to determine the substances, which could be evolved during the thermal decomposition of the PU with fire retardant system. The results indicated a reduction in heat release rate (HRR), maximum average rate of heat emission (MAHRE), the total heat release (THR) as well as the total smoke release (TSR), and maximum specific optical density (Ds_max_) compared to the polyurethane with commercial fire retardant, namely ammonium polyphosphate (APP). A significantly improvement, especially in smoke suppression, suggested that HGO_A_ system may be a candidate as a fire retardant to reduce the flammability of PU foams.

## 1. Introduction

Polymers containing a urethane bond in the structure of macromolecules are called polyurethanes (PU). They are obtained as a result of the gradual polyaddition reaction of compounds containing two or more isocyanate groups in the molecule with those containing two or more hydroxyl or amino groups [1]. Polyurethanes are composed of rigid and flexible segments. The rigid segments are made of a diisocyanate and a chain extender, and the flexible segments are polyol. After the PUR macromolecule formation, the rigid segments are joined together by hydrogen bonds, which leads to the formation of hard and soft domains [2].

Polyurethane foams have the highest share in the market of polyurethane materials, among which there are distinguish flexible, semi-rigid, and rigid foams. Rigid foams account for 23% of the total production of polyurethane materials [3]. They are mainly used to fill spaces in building structures, in window frames and doors, for the production of structural insulation boards, for insulation of tanks and pipelines, as well as for thermal and acoustic insulation and reinforcement of floor and ceiling structures [4,5,6].

PUR foams are flammable due to their organic nature and a very well developed surface [7]. Polyurethanes decompose at temperatures above 200 °C, which is accompanied by the release of toxic substances. The volatile products of polyurethane pyrolysis are phenylisocyanate and p-tolueneisocyanate, as well as isoquinoline, o-benzodinitrile, and hydrocarbons (toluene, xylene, benzene, biphenyl, naphthalene, carbazole, dimethylbiphenyl). PU decomposition products also include nitriles and nitrogen oxides. The type and content of thermal decomposition and burning products depend on the chemical structure of polyurethane [8].

The fire retardancy of polyurethane foams can be increased as a result of incorporating reactive compounds into their structure, adding non-reactive compounds (fire retardants) and coating the product with non-flammable insulation coatings. Among the numerous methods of limiting the flammability of foams, the use of fire retardants (FR) is the most significant. Their task is not only to reduce flammability and smoke emission or to limit the toxicity of gaseous burning products, but also not to affect or even improve the functional properties of the material and remain in the foamed polyurethane during long-term use [7,8,9].

Among the wide range of flame retardants available for the polyurethane foams, chlorine and phosphorus compounds such as TCEP (tris (1,3-dichloropropyl) phosphate) and TCPP (tris (2-chloroisopropyl) phosphate) have been widely used due to low price and good efficiency. Halogen compounds are still a popular group of flame retardants used in rigid polyurethane foams, but they do not meet the present requirements for fire retardants. The addition of halogen FR causes the emission of highly toxic gases that are hazardous to human health and even life [8,9,10]. Therefore, intensive research into halogen-free flame retardants has begun. Compounds based on phosphorus, nitrogen, silica, and boron are gradually replacing halogenated flame retardants [11,12,13,14]. To reduce the flammability of foams, phosphorus-containing compounds, such as phosphates, red phosphorus, phosphites, phosphonates, alkyl phosphamides, melamine phosphates, triphenyl phosphate, and triethyl phosphate are used. One of the most frequently used phosphorus FR is ammonium polyphosphate (APP) [9]. Metal hydroxides such as aluminum trihydroxide (ATH) or magnesium dihydroxide (MH) and expandable graphite (EG) were also widely studied as a promising environmentally friendly flame retardant with good results. Nitrogen and phosphorus flame retardants, known as intumescent flame retardants (IFRs), are considered an effective way to improve the fire behavior of polyurethanes. However, additive flame retardants have certain disadvantages, such as phase separation due to high loads, loss of homogeneity, high viscosity of compositions, etc. [15,16,17]. The addition of a large amount of fire retardant, to achieve the required level of flame retardancy, causes numerous processing problems and significantly affects the physical and mechanical properties of the materials. Increasing fire resistance can also be obtained by using several flame retardants acting synergistically [8,9].

The goal of this study was to investigate the influence of a non-halogen fire retardant system, consisting of histidine and graphene oxide (HGO_A_), on thermal stability and flammability of rigid polyurethane foams. The flame retardancy as well as smoke emission of the PU containing from 10 to 30 wt.% of FR system were evaluated by cone calorimetry and smoke density chamber measurements. The impact of the HGO_A_ was confirmed by a microstructure analysis of the samples before and after CC tests. Furthermore, TGA combined with FT-IR was conducted to determine the substances, which could be evolved during the thermal decomposition of the PU/FR. The properties of prepared materials were compared to the polyurethane with commercial FR, namely ammonium polyphosphate (PU/APP).

## 2. Materials and Methods

Polyether polyols Petol 400 and Petol 480 were supplied from Minova Ekochem (Siemianowice Śląskie, Poland). The primary properties of Petol 400 and Petol 480 were as follows: hydroxyl value 385 ± 25 and 475 ± 15 mgKOH/g, viscosity (temperature 25 °C) 330–440 and 6500–900 mPa·s, respectively. Furthermore, 33.3% triethylenediamine in propylene glycol PC CAT TD-33 and 1,3,5-tris [3- (dimethylamino) propyl] hexahydro-1,3,5-triazine PC41, both supplied by Minova Ekochem, were used as catalysts. Silicone foam stabilizer Niax silicone L-6900 was purchased from Minova Ekochem. Pentan was used as a blowing agent. The isocyanate component was polymeric MDI commercially traded as Ongronat 2100, supplied by Minova Ekochem.

Histidine was purchased from Apollo Scientific (>99%, Cheshire, UK). The structure of H is present in Figure 1a.

The other chemicals were used for graphene oxide synthesis and modification. The flake graphite was purchased from Asbury Carbons (Asbury, NJ, USA). Sulphuric acid (H_2_SO_4_), potassium permanganate (KMnO_4_), potassium nitrate (KNO_3_), perhydrol (30% H_2_O_2_), and ethanol (96%) were purchased from Chempur (Piekary Śląskie, Poland). All these reagents were pure for analysis. Additionally, 98% APTES (3-Aminopropyltriethoxysilane) was delivered from Alfa Aesar (Haverhill, MA, USA). The APTES structure is present in Figure 1b.

Moreover, ammonium polyphosphate (APP), with the particle size below 15 µm, was used to manufacture a reference PU/FR.

Graphene oxide (GO) was prepared by modified Hummers method [18]; 5 g of flake graphite was poured into beaker with solution of 3.25 g of KNO_3_ in 350 mL of H_2_SO_4_ and thoroughly mixed with mechanical stirrer (IKA RW16 basic, IKA® Werke GmbH & Co. KG, Staufen, Germany). The beaker with reagents was then put into water/ice bath, and 30 g of KMnO_4_ was gradually added into the mixture. After last portion of oxidant was added, the beaker was taken from the bath and kept for 3 h at 30–35 °C with continuous stirring. Then it was left at room temperature for two days. After that time, the deionized water was carefully added to the mixture, so its temperature had not exceeded 35 °C. In the next step, the acid–graphite oxide mixture was heated to 95 °C, while stirred vigorously, and kept under these conditions for 15 min. The heater was then turned off and the beaker was allowed to cool slowly. When temperature fell under 40 °C, 280 mL of deionized water and 20 mL of H_2_O_2_ were added. The graphite oxide mixture was purified by centrifugation (Thermo Lynx 4000, Osterode, Germany). It was then sonicated for 1 h with ultrasonic processor (Sonics and Materials INC, VCX750, Newtown, CT, USA) to exfoliate graphite oxide, thus obtaining graphene oxide. Then the graphene oxide suspension was deprived of water using spray dryer (BUCHI, B-290, Uster, Switzerland) to prepare the powder for further modification. The portion of dried GO in quantity 1.25 g was sonicated for 0.5 h in 50 mL 90% ethanol solution. After that time, 1 mL of APTES was added and the solution was mixed with magnetic stirrer (Heidolph MR Hei-Standard, Schwabach, Germany) for 0.5 h. The modified graphene oxide (GO_A_) was washed three times with the deionized water and then freeze-dried for 24 h (Martin Christ, BETA 1-8 LD plus, Osterode am Harz, Germany).

PU foams were prepared via a one-step procedure with a modification, consisting of the preparation of a polyol masterbatch. The formulations are listed in Table 1. Firstly, the dispersion of GO_A_ in the polyols was carried out using a high-speed mechanical stirrer (proLAB 075, GlobimiX, Ząbkowice Śląskie, Poland) with the following rotational speeds: 3000, 5000, and 10,000 rpm applied for approximately 300, 240, and 30 s, respectively. Next, the composition was subjected to homogenization using an ultrasonic disperser (Q700, Qsonica, Newtown, CT, USA). The amplitude of the process was 50% with a dispersion time of about 20 min. During the mixing and homogenization processes, the temperature of the mixture was controlled, and the system was cooled in an ice-water bath so that its temperature did not exceed 50 °C. Then, a histidine was added, and the mixture was stirred using the following speeds and times: 3000 rpm for 180 s, 5000 rpm for 120 s, and 10,000 rpm for 60 s. In the next stage, the catalysts and the surfactant were introduced and re-subjected to the mixing process. Subsequently, the blowing agent was added, and the pre-mix was mixed at a speed of 3000 rpm for 10 s. In the final stage, the isocyanate component was introduced to the system, mixed at 3000 rpm over 10 s and poured into an open mold. Process times for PU/APP and PU/HGO_A_ are presented in Table 2. After foaming, the PU foams were removed from the mold, cured at 70 °C for 30 min., and conditioned at ambient temperature for two weeks. Then, the PUs was cut into samples according to the test standards.

Scanning electron microscope SU8010 (Hitachi High-technology Corporation, Hagawa, Japan) was used to determine the morphology of the fire retardant system components, polyurethane foams, as well as the residues after cone calorimetry measurement. To improve the conductivity, components and PU foams were gold-coated using a Q150T ES device (Quorum Technologies, East Sussex, UK). Imaging was performed with secondary electrons at an acceleration voltage of 10 kV and at magnifications 40× in the case of foams or 1000 and 2000× for FR system components. SEM images were used to calculate the mean equivalent diameter and aspect ratio of pores (N ≥ 500 for each foam variant). Point elemental analysis was carried out using the Thermo Scientific NORAN System 7 equipped with electrically cooled Silicon Drift Detector EDS detector (Thermo Scientific UltraDry, Waltham, MA, USA).

The thermogravimetric analysis of the fire retardant components and polyurethane foams was done using a TGA Q500 (TA Instruments, New Castle, DE, USA); 10 ± 0.5 mg samples were tested in air with flowing gas at a rate of 10 mL/min in the chamber and 90 mL/min in the oven. Materials were heated in the temperature range from ambient to 1000 °C at a rate of 10 °C/min. The obtained data were analyzed using Universal Analysis 2000 ver.4.7A software (TA Instruments, New Castle, DE, USA).

The chemical structure of the rigid polyurethane foams was determined based on infrared absorption spectra using a Nicolet 6700 spectrophotometer (Thermo Electron Corporation, Waltham, MA, USA) with an attenuated total reflection (ATR) accessory. Samples 4–5 mm thick were used for the tests. Each sample was scanned 64 times in the wavenumber range 4000–400 cm^−1^. The results were analyzed using the OMNIC 8.2.0 software (Thermo Fisher Scientific, Waltham, MA, USA).

The apparent density was determined according to the standard [19]. Samples with dimensions 5 × 5 × 5 cm were measured with an accuracy of 0.1 mm and weighed in the air with an accuracy of 0.001 g using a weighing machine WPA 180/C/1 (Radwag, Radom, Poland).

The friability of rigid PUR foams was according to the standard [20]. Twelve cubic cubes with a side of 25 mm were cut from each material. The weighed cubes were placed in an oak rotating drum, together with 24 oak cubes, and made to rotate. The test was carried out at a rotational speed of 60 rpm for 10 min. Next, the samples were cleaned of dust and weighed again. The friability was determined according to the Equation (1):

(1)
$$K=\left(\frac{m_{1}-m_{2}}{m_{1}}\right)\times 100\%$$

where: *m*_1_—a mass of samples before testing, *m*_2_—a mass of samples after the test.

The differential scanning calorimetry (DSC) was made to determine temperatures and thermal effects of phase changes. The study was performed using a DSC Q1000 device (TA Instruments, New Castle, DE, USA). Samples of the foams were placed in sealed aluminum crucibles, which were cooled down to −90 °C and heated to 170 °C at a rate of 10 °C/min. The analysis was performed in the heating/cooling/heating cycles. The results were analyzed using the Universal Analysis 2000 ver.4.5A software (TA Instruments, New Castle, DE, USA).

Fire behavior was assessed using a cone calorimeter (Fire Testing Technology, East Grinstead, UK). The samples (100 × 100 × 25 ± 1 mm) were tested at an applied heat flux horizontally of 35 kW/m^2^, in conformity with ISO 5660 standard [21]. Separation space between samples and the heater was set at 25 mm. The residues were photographed using a digital camera EOS 400 D (Canon Inc., Tokyo, Japan).

The optical density of smoke was assessed using a smoke density chamber (Fire Testing Technology). The samples (75 × 75 × 10 ± 1 mm) were exposed to a heat flux of 25 kW/m^2^ without the application of pilot flame, in conformity with ISO 5659-2 [22] standard. The values, as in the case of the CC tests, are the average obtained for three samples from each series.

Analyses of gases evolved during the TGA experiments were performed using a FTIR Nicolet 6700 spectrometer (Thermo Electron Corporation, Waltham, MA, USA) coupled to TGA Q500 (TA Instruments, New Castle, DE, USA), and comprised of 64 scans per minute. The samples (10 mg ± 0.5) were heated in air from room temperature to 860 °C at a rate of 20 °C/min. To reduce the possibility of the evolved products condensation along the transfer line, the FT-IR gas cell was held at 240 °C and the temperature of the transfer line was set to 250 °C. The analyses were performed in spectral range 400–4000 cm^−1^.

## 3. Results

### 3.1. Fire Retardant’s Characteristic

SEM images were utilized to characterize the morphology of fire retardant system components.

As illustrated in Figure 2a,b, the particle diameter of histidine varies from several dozen nm to 50 µm. Moreover, the H particles tended to aggregate with the formation of increased size aggregates. Most of the histidine particles visible on the SEM images are agglomerates of a very large number of particles, possessing a super-microporous structure [23]. Moreover, SEM images allow observing that the carried out modification increased the GO_A_ size from nano to micrometers. The dried graphene oxide flakes were deformed, losing their flat shape for a three-dimensional structure.

The quality of the dispersion of solid additives has a significant impact on the course of the foaming process and the quality of the cell structure of the obtained materials. Filler particles tend to agglomerate, forming large aggregates in the PU matrix. Moreover, too large particles can disturb the foaming process leading to the formation of materials with a heterogeneous structure. On the other hand, the large filler agglomerates can lead to gravitational drainage of cells in the initial foaming stage.

A subsequent research technique used to characterize the properties of the components of the FR system was thermogravimetric analysis. Figure 3 shows TG and DTG curves of H and GO_A_, while some data are listed in Table 3.

Contrary to histidine, the graphene oxide modified with the APTEST method contained a significant amount of water, or volatile products, reaching even a dozen percent. Such a high content was the reason for a low temperature value corresponding to 5% weight loss (T_5%_). The T_5%_ temperature for GO_A_ was only 57 °C, while for histidine it was 262 °C. It cannot be ruled out that this also had an impact on the T_50%_, which in the case of H and GO_A_ was 540 and 437 °C, respectively. Notably, the residue at 950 °C in the test conducted in aerobic conditions for graphene oxide was as high as 16%.

### 3.2. Characterization of the Polyurethane Foams

The FT-IR spectra of PU/APP indicate the presence of groups, characteristic for polyurethanes, which confirms the completion of rigid polyurethane foam synthesis.

The presence of -N-H groups in the urethane moiety was confirmed by a signal with a maximum at 3304 cm^−1^ (stretching vibrations of N-H bond, symmetric and asymmetric) and a signal with a maximum at 1518–1517 cm^−1^ (deformation vibrations of N-H). The absorption bands with maxima at 2978–2972 cm^−1^, 2927–2925 cm^−1^, and 2870 cm^−1^ (Table 4) originate from the asymmetric and symmetric stretching vibrations of the C-H bonds in the -CH_2_- and -CH_3_ groups. A signal with a maximum at 2279–2275 cm^−1^ indicates the presence of N=C=O moieties derived from unreacted isocyanate, which is related to the used the isocyanate index value of 110 (Figure 4).

The presence of carbonyl groups in the urethane moiety was confirmed by the band with a maximum at 1708–1707 cm^−1^ (stretching vibration of C=O bond). A signal at a wavenumber of 1595 cm^−1^ indicates the presence of aromatic rings derived from the used isocyanate. The presence of isocyanate trimerization products is indicated by the 1411 cm^−1^ signal. An absorption band with a maximum at 1219 cm^−1^ indicates stretching vibration of C-N groups in polyurethanes. Multiplet signals in the wavenumber range of 1150–1000 cm^−1^ are related to the vibrations in elastic segments [2,5,24]. The introduction of the APP flame retardant resulted in the broadening of the absorption bands in the wavenumber range 400–600 cm^−1^ related to the bending vibrations of the C-H groups.

The interpretation of the absorption bands of the FT-IR spectra for PU/HGO_A_ foams is analogous to those obtained for the APP modified foam. For these materials, an additional signal was observed in the range of bending vibrations of the C-H bonds at the value of 623 cm^−1^, which can be assigned to the presence of histidine. The intensity of this signal increases with the increasing content of the HGO_A_.

### 3.3. Physico-Mechanical Properties and Microstructure Analysis of Polyurethane Foams

The microstructure is one of the most notable factors that may affect the properties of PU foams. The structure, essentially cell size and type, depends on the parameters of the process, especially viscosity of mixture, pressure, and temperature during the foaming. In turn, morphological parameters, including cell size, wall, and ribs thickness, have a meaningful impact on the physico-mechanical properties of PU foams [23,25]. The microstructures of the investigated rigid polyurethane foams are presented in Figure 5, while the main characteristics of the cellular structure are listed in Table 5.

The shape of cells in the case of the PU/APP was a typically closed polyhedron with an equivalent diameter (D) ranging from 350 to 750 μm. For both APP and HGO_A_, the growing share of FRs in foams led to the creation of a higher amount of open cells with thicker walls, resulting from FR particles built into the polyurethane ribs [25]. Moreover, the non-linear increase in the D values (Table 5) was also observed. Notably, PU foams with APP had a smaller cell diameter in comparison with PU/HGO_A_, as well as lower aspect ratio (Ar) values, suggesting that the shape of the PU/APP cells was less elongated. For all foams, the increase in FR amount contributed to the decrease in Ar values. It is worth emphasizing that, along with the amount of FRs, the range of diameter sizes has also increased, as indicated by the graphs in Figure 5 (X-axis). Apparently, the particles of fire retardants acted as nucleation sites, causing the simultaneous formation of cells with a smaller diameter [26,27]. As demonstrated in the previous studies, the solid particles are likely to influence the rheology around the growing air bubbles and reduce the nucleation energy, thus inducing the change of nucleation type from homogenous to heterogeneous. The low nucleation barrier facilitates the extensive formation of smaller cells, which later coalesce into larger units [28,29].

One of the most significant parameters, determining the use of PU foams as a thermal insulation material, is the apparent density, which indirectly influences mechanical properties, such as compressive strength or fragility [30]. The addition of flame retardants caused an increase of the density along with their increasing share as an effect of heterogeneous nucleation and a growing number of small size cells [23]. The apparent density of the investigated PU/FRs foams is within 30–45 kg/m^3^, and it is approximately 10 kg/m^3^ lower for samples with HGO_A_ compared to PU/APP with the same amount of FR. This is probably due to the smaller APP grain size compared to the H agglomerates and the higher proportion of small cell sizes for PU/APP. In the case of all materials, the growth in friability, contributed by the increase in the share of FRs was observed; however, much higher values were recorded for PUR modified with HGO_A_ system. The used of 30 wt.% of histidine and graphene oxide reduced the mechanical properties of rigid PU foams by one third. The reason may be the large size of the histidine agglomerates, which were not damaged despite the applied two-stage homogenization process. The large size of H compared to other FR contributed to the creation of a large number of significant size cells, weakening the structure of the foams.

### 3.4. Thermal Properties

The thermal stability of PU is related largely to structure, the equivalent ratio of functional groups of the hard and soft segment, as well as the degree of phase separation [31]. The thermo-oxidative properties of polyurethane foams and PU/FRs were studied in the air, and the results are summarized in Table 6 and Figure 6.

All investigated polyurethane foams remained thermally stable up to temperature 220–256 °C (T_5%_). Then, a rapid decomposition process caused 50% of weight loss (T_50%_) to occur at temperature range 314–396 and 313–328 in the case of PU/APP and PU/HGOA, respectively. The initial weight loss was associated with the dissociation of biuret and allophanate, as well as the evaporation of water [32]. Flame retardant polyurethane foams presented three major thermal transitions, approximately 280, 520, and 780 °C. The first wide peak (Figure 6) corresponds to the decomposition of the urethane to obtain isocyanate monomers and polyols segments, which degrade next [33]. The second thermal event is due to decomposition of transient char, appearing in an oxidative atmosphere. The last step, occurring only in the case of PU/APP, is a result of further oxidation of char, which decreased its quality as the heat supply growth [32].

Moreover, the intensity of degradation (Table 6) shows that the decomposition rate of PU/HGO_A_ was lower compared to PU/APP. It was probably due to the formation of more stable residue than for PU with the same amount of commercial FR. As a result, the residual mass of foams modified with developed FR system was similar or higher than that of APP. However, because of the oxygen conditions, as well as the small weight of samples, the yield of char in both cases was relatively small, and no linear dependencies resulting from the share of FR were observed. Similar observations, describing the thermal stability of polymers modified by the newly developed flame retardant assessed based on TG analysis under aerobic conditions, can be found in the literature [34]. It can be concluded that the presence of histidine and modified graphene oxide led to the quicker start of the PU’s decomposition than APP; however, the created char was characterized by a lower intensity of decomposition (better thermo-oxidative stability).

The curves obtained by the first heating cycle (C1) of the materials differ from the curves obtained by heating the samples a second time (C2) (Figure 7). The C1 thermograms show two endothermic peaks: one in the range −80 to −40 °C, related to the soft phase changes of polyurethane (ΔH_SS_), and the other in the range of 50 to 160 °C, related to the order–disorder changes in the hard phase of polyurethane (ΔH_HS_) [35]. C2 thermograms obtained in the second heating cycle were used to determine the hard phase glass transition temperature (Tg_HS_) (Figure 8). Due to the small thermal effects accompanying the transition in the soft phase in the second heating cycle C2 and the presence of an endothermic signal on the C1 curve, it was not possible to determine the glass transition temperature in the soft phase of the materials. The results of the thermogram analysis indicate that the glass transition temperature of the foams made with APP is higher compared to the glass transition temperature of the foams made with HGO_A_, which indicates that the introduction of HGO_A_ into the polyurethane system increased the mobility of the chains forming the hard phase. The use of APP or HGO_A_ flame retardants did not result in any effect on Tg_HS_ of the samples.

### 3.5. Fire Behavior

CC tests simulate a developing fire scenario with the use of a small sample. Since the tests are carried out in external heat flux, they are often used to estimate polymers’ forced burning fire performance [36]. The common parameters of interest in CC measurements are the time to ignition (TTI), the maximum peak heat release rate (pHRR) for the HRR curve, maximum average rate of heat emission (MAHRE), the total heat release (THR), as well as the total smoke release (TSR).

Heat release rate versus time curves, shown in Figure 9, presents the typical burning behavior of polyurethane foams. This type of materials exposed to the external heat flux ignites within a few seconds because of its low thermal inertia, caused by the polymer’s low thermal conductivity [37]. Rapid heating of the cell walls changes the absorbing layer of radiation into liquid pyrolysis products, leading to the fast development of burning [38]. As can be seen in Figure 9, the ignition was followed by the maximum peak, yielding the pHRR and plateau-like behavior, with less intense burning in the case of samples with HGO_A_. The intensity of burning depended mostly on the type of FR. The detailed data are shown in Table 7.

Since the PU foams are characterized by cellular structure and low thermal inertia [39,40], the time to ignition (TT) of all samples was 6 s or less. Reduced thermal inertia of foams with lower density may lead to decreased TTI values [38], as in the case of PU/10HGO_A_; however, no trend was observed. PHRR were similar within APP and HGO_A_ series independently of FR amount; nevertheless, insignificantly lower values were obtained for PU with HGO_A_. The differences between the pHRR for PU with 10%, 20%, and 30% of APP and HGO_A_ were 21%, 24%, and 21%, respectively. A similar trend was observed for MARHE, and the lowest results were recorded for samples with the lowest amount of additives.

For most samples, the THR reached 19 kJ/m^2^ and decreased to 16–17 kJ/m^2^ for the PU with the highest share of HGO_A_, which may result from the reduced density of this samples. The lower the mass of samples, the lower amount of pyrolysis products and heat that may be released [37]. However, a decrease of fire load was probably caused by a char’s formation, as confirmed by the amount of residue and samples’ photographs after the CC measurements shown in Table 7 and Figure 10, respectively. The char yield in PU with APP changed negligibly with a share of FRs, but the standard deviation suggests that values are not the same. In the case of PU/HGO_A_, greater differences between the values were observed, ranging from 19% to 28%. The highest residue was obtained for PU/20HGO_A_, and results are following the residual mass obtained from TG analysis.

Photographs of residues after CC tests are presented in Figure 10, while microstructure images in Figure 11. The differences in the residues’ quality on a macroscopic level are very expressive, although the SEM investigation showed that both chars were closed and compact in the outer part as well as porous inside. Since the char of PU/APP’s surface was thin and not sufficiently large, it provided only partial protection. In turn, PU/HGO_A_ exhibited quite a lot of holes. The chemical composition designated that residues comprised of C, O, N, and P elements in the case of PU modified with a commercial fire retardant. The results suggested that the FRs could decompose to the products, actively participating in the crosslinking process, such as P_2_O_7_, which affects a more compact char with better mechanical performance [41]. Moreover, the Al elements for PU/30APP, derived from the sample’s aluminum cover, confirm that the char layer was skinny.

### 3.6. Smoke Emission

Toxic, high-temperature smoke is the major cause of fire-related deaths in enclosed spaces. Typical PU foams are easily ignited by a small flame and burn rapidly, emitting a high amount of smoke [36]. Concerning the smoke production, total smoke release (TSR) curves are presented in Figure 12, and the related data are summarized in Table 8.

The total smoke release indicates cumulative smoke amount produced per unit area of the sample [36]. As shown in Figure 8, PU/HGO_A_ samples exhibited a significant decrease in TSR during the entire burning process than the foams with commercial FR. Importantly, the amount of emitted fumes successively decreased with the increase in the amount of HGO_A_, contrary to APP. The TSR value of PU/30HGO_A_ was 295 m^2^/m^2^, representing a 58% reduction compared to the PU/30APP (706 m^2^/m^2^). Similar results were recorded for specific extinction area (SEA), corresponding to the surface of light-absorbing particles present in smoke produced from 1 kg of material. In particular, the PU/20HGO_A_ with SEA of 359 m^2^/kg was 49% lower than of sample with the same share of APP, a clear indication of the excellent smoke suppression effects investigated FR system.

Subsequently, smoke density chamber tests were carried out to evaluate whether or not the smoke suppressant mode coincides with the previous assumption. Similarly to the above results, along with the increase in the APP share in the foam, the maximum specific optical density (Ds_max_) as well as VOF 4, informing how much smoke is released during the first 4 minutes, needed to carry out an effective evacuation action, were increased. One of the reasons for the suppression of smoke is the formation of char and incorporation of incomplete burning products into the condensed phase. The obtained results correlate with the yield of residues, the value of which is presented in Table 6. 

### 3.7. Characterization of Decomposition Products

The combination of the TG analyzer and FTIR spectrometer creates a valuable opportunity for understanding the thermal decomposition pathways due to the detailed analysis of the evolved volatile products. The 3D TGA/FT-IR spectra of the gas phase and the spectra of volatilized products released at the maximum decomposition rate of tested materials were presented in Figure 13 and Figure 14, respectively.

The obtained results indicate two stages of thermal degradation of PU/30HGO_A_ (in 16.01 and 27.97 min) and three stages of thermal degradation PU/30APP (in 15.98, 27.17, and 37.13 min). Peaks observed in the first stage of material degradation, in the range of 4000–3500 cm^−1^, are related to the O-H stretching vibrations from water as well as N-H stretching vibrations. Signals observed in the range 3020–2820 cm^−1^ are connected to the C-H stretching vibrations from CH_2_ groups. In turn, peaks at 2358, 2322, and 668 cm^−1^ are related to CO_2_, which is a product of urethane groups decomposition. Moreover, low intensity peaks at wavenumbers 2181 and 2099 cm^−1^, related to CO, were observed. The peak at 1507 cm^−1^ is due to the stretching vibrations of the aromatic ring formed upon pyrolysis. The peak at 1456 cm^−1^ was assigned to the -CH_2_- bending vibrations. The further peaks located at 1751 and 1116 cm^−1^ are related to the C=O and C–O–C groups, respectively [42,43]. For the PU/30APP material, additional signals were observed at the values of 966 and 930 cm^−1^, which can be assigned to the presence of NH_3_. In the second stage of material degradation, signals related to CO_2_ and CO release for both materials were observed. For the PU/30APP material, the third degradation stage was observed accompanied by CO_2_ and CO release. The observed pathways of thermal degradation may result from the formation of char, acting as an effective physical barrier limiting the release of pyrolysis products in the case of both materials. However, observed degradation course suggests that the char formed in the pyrolysis of PU/30APP undergoes further decomposition in the third stage, differently from the PU/30HGO_A_. This conclusion is corroborated by the higher share of carbon in the combustion residue of PU/30HGO_A_ (Figure 9).

### 3.8. Fire Retardant Mechanism

Histidine is an amino acid; this means that it has an amine group and an organic acid group. In mechanism I, it was shown that the amine group detaches from the particle forming gaseous ammonium and nitrogen oxides (especially under higher temperatures and with access to atmospheric oxygen). It is probable that during the decomposition, some amount of hydrogen cyanide may form [44,45]. The acidic group typically decomposes into water and carbon oxides, both non-flammable gases (water also providing the heat sink effect) that work to the advantage of char formation. In mechanism II, according to literature reports, histidine may also undergo the inner cyclization process, forming a structure consisting of two rings: 2-amino-2,4-cyclopentadien-1-one and imidazole, which co-create the char structure (Figure 15). The generation of a high amount of non-flammable gaseous products combined with a thick and strong char structure helps protect the polyurethane from external heat.

## 4. Conclusions

The impact of the mixture of histidine and graphene oxide on the burning behavior of rigid polyurethane foams was assessed, and the results were compared to commercial fire retardant.

Apart from a less homogeneous structure of all samples, noticeable voids in foams structure were observed, while FR’s particles built into the polyurethane ribs caused a thickening of cell walls. PU/HGO_A_ were characterized by lower density as well as increased friability compared to PU/APP. TG measurements exhibited quite similar oxidative-pyrolysis temperatures for PU/APP and PU/HGO_A_; however, PU/HGO_A_ char was more stable in higher temperature, contributing to their improved fire performance.

Under forced-flaming conditions, all the samples ignited almost immediately; but pHRR, MARHE, and THR were lower for PU with HGO_A_ system than for commercially available FR. A decrease in fire spread and fire load were caused by a char’s formation, higher in the case of PU/HGO_A_. The increase in charring mostly affected significant higher suppression of smoke.

## Figures and Tables

**Figure 1 materials-14-01184-f001:**
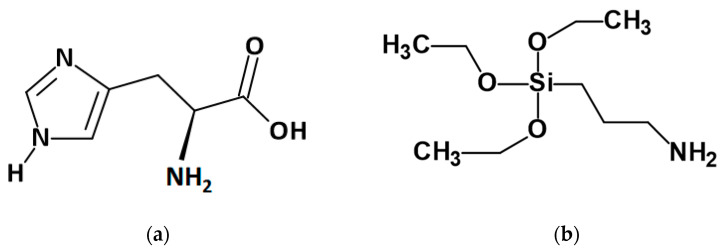
The chemical structure of histidine (**a**) and 3-Aminopropyltriethoxysilane (**b**).

**Figure 2 materials-14-01184-f002:**
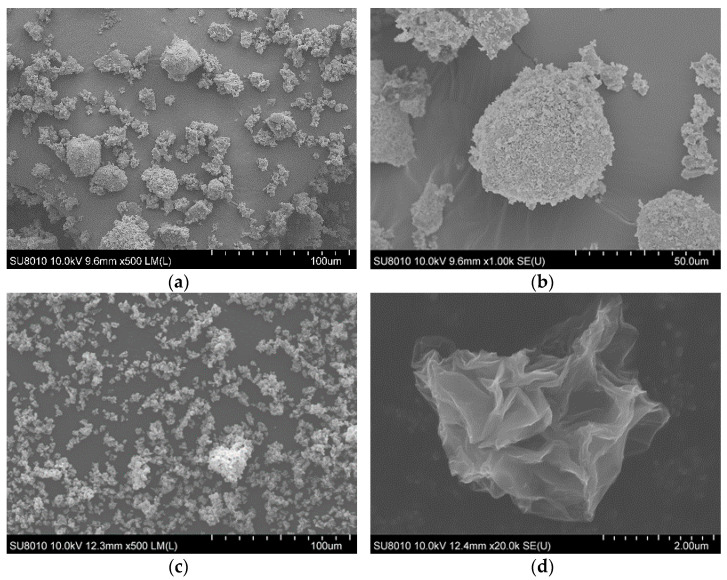
SEM images of H (**a**,**b**) and GO_A_ (**c**,**d**).

**Figure 3 materials-14-01184-f003:**
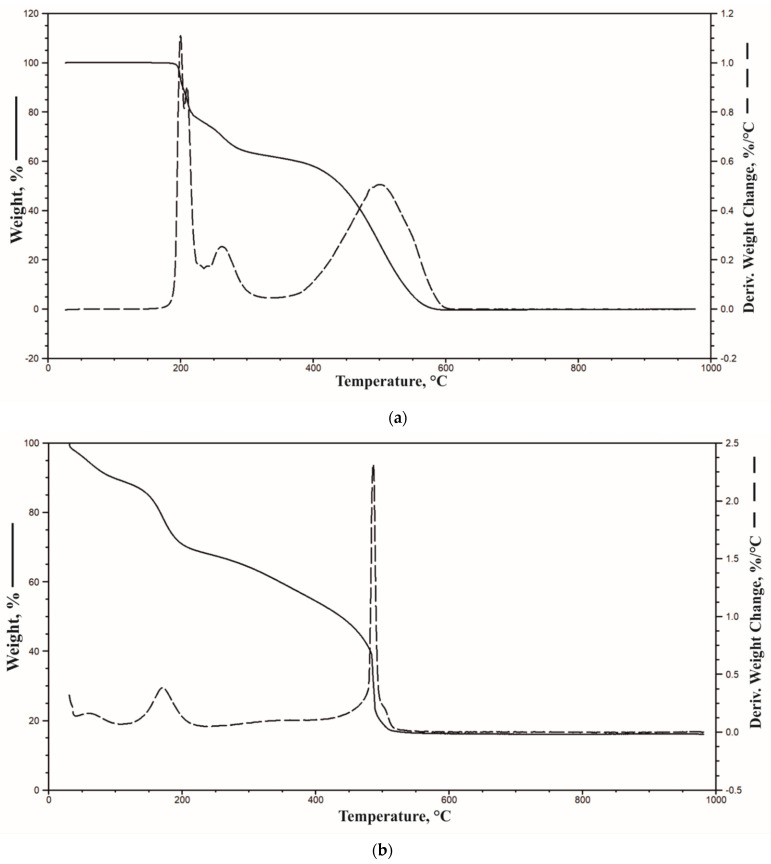
TG and DTG curves of H (**a**) and GO_A_ (**b**) in air.

**Figure 4 materials-14-01184-f004:**
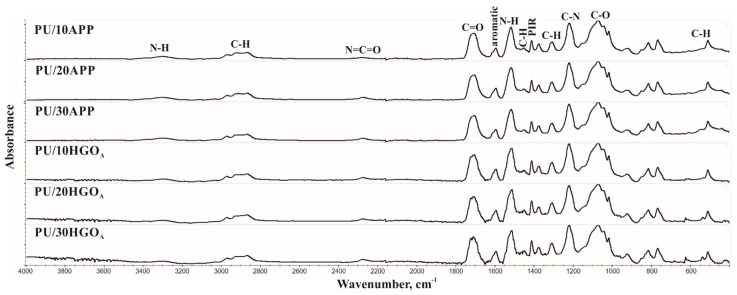
FT-IR spectra of PU/APP and PU/HGO_A_ foams.

**Figure 5 materials-14-01184-f005:**
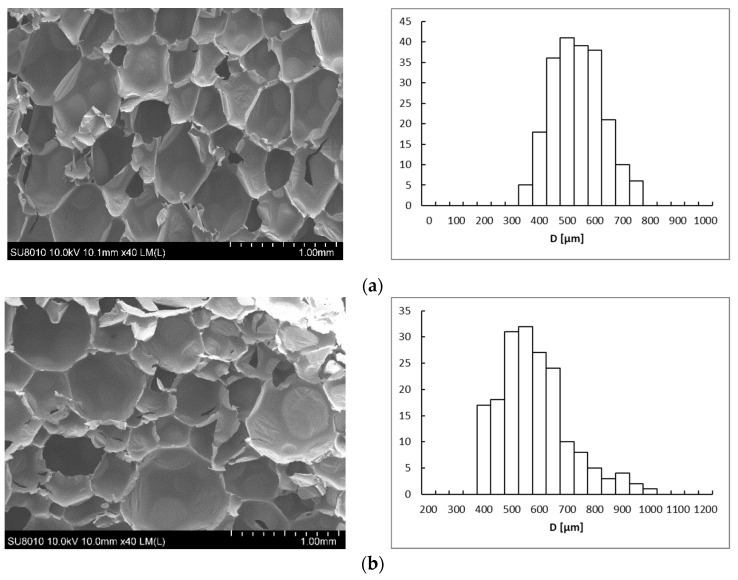
SEM images of breakthroughs of PU/10APP (**a**), PU/20APP (**b**), PU/30APP (**c**), PU/10HGO_A_ (**d**), PU/20HGOA (**e**), and PU/30HGO_A_ (**f**).

**Figure 6 materials-14-01184-f006:**
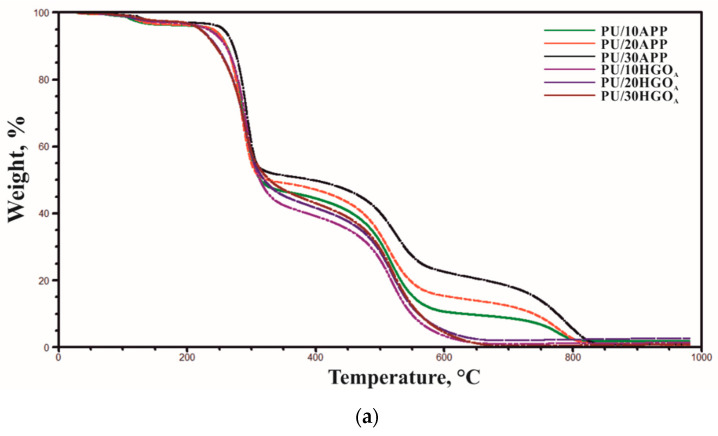
TG (**a**) and DTG (**b**) curves of PU/APP and PU/HGO_A_ in air.

**Figure 7 materials-14-01184-f007:**
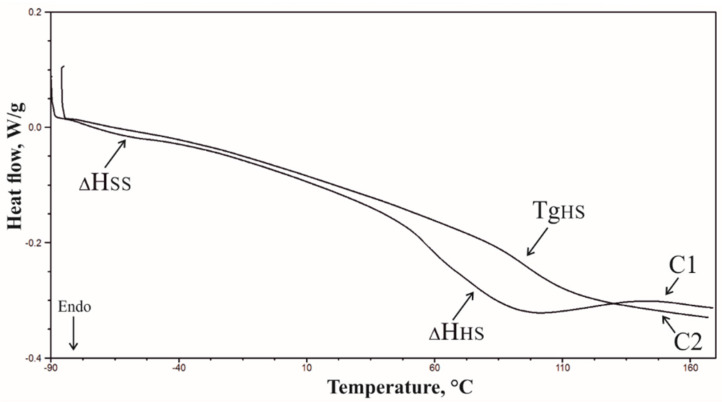
DSC thermogram of the PU/20 HGO_A_ material.

**Figure 8 materials-14-01184-f008:**
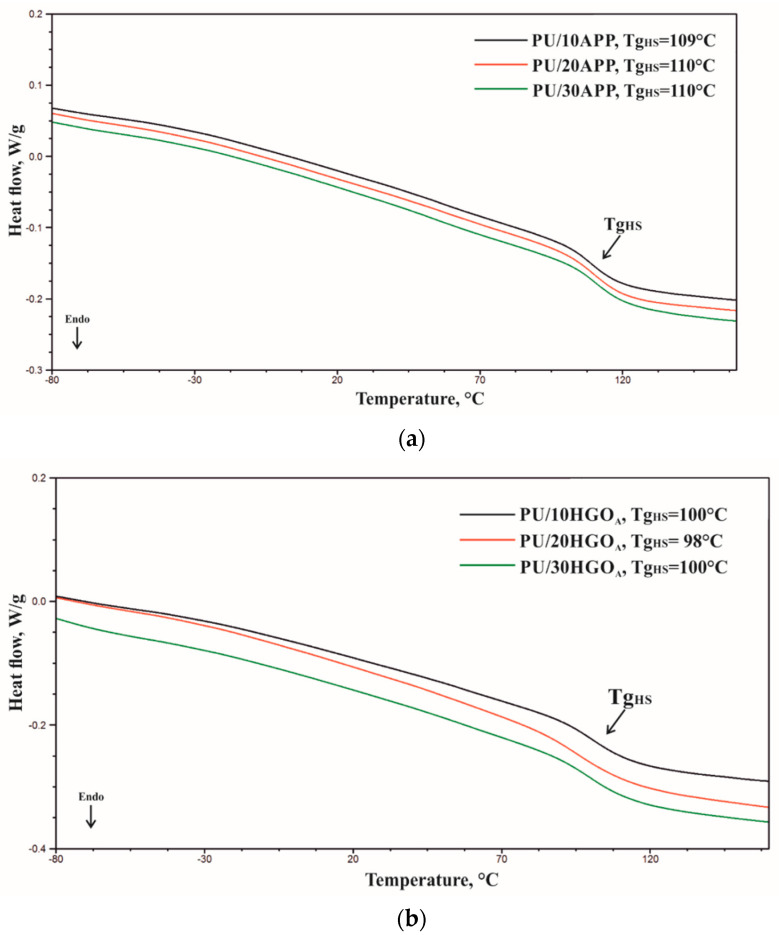
Tg_HS_ of materials (**a**) PU/APP (**b**) PU/ HGO_A_.

**Figure 9 materials-14-01184-f009:**
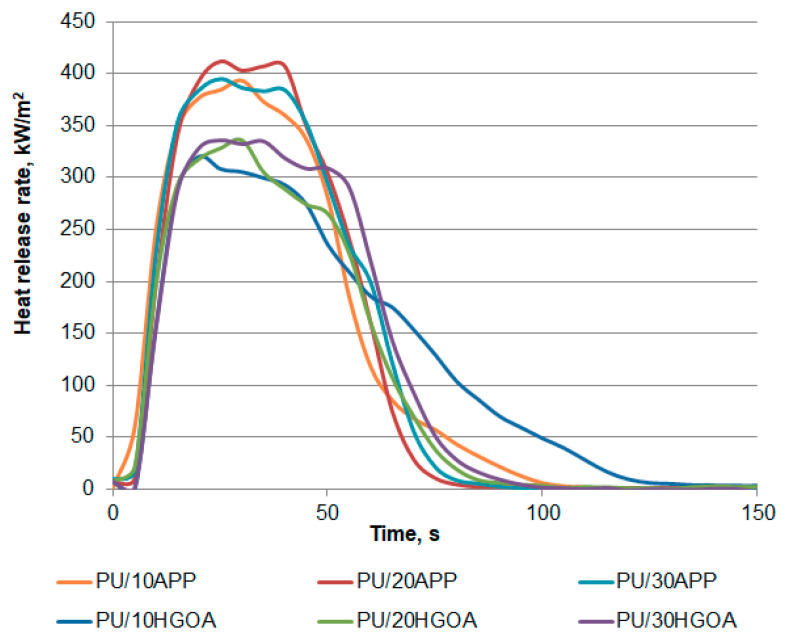
Representative curves of heat release rate of polyurethane foams with APP and HGO_A_ system.

**Figure 10 materials-14-01184-f010:**
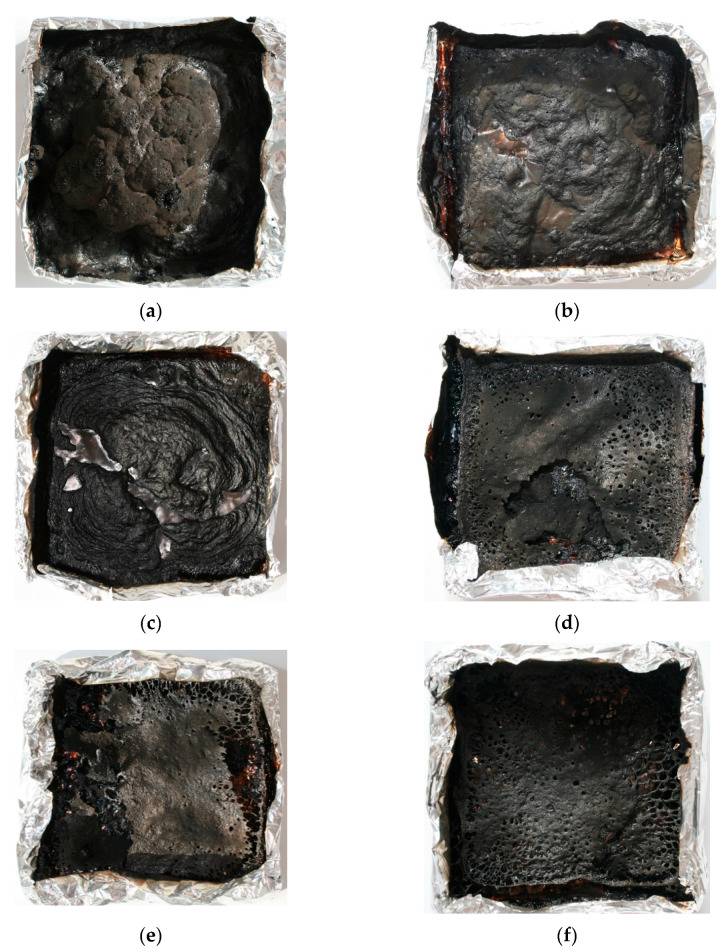
Photographs of the PU/10APP (**a**) PU/20APP (**b**) PU/30APP (**c**) PU/10HGO_A_ (**d**) PU/20HGO_A_ (**e**) and PU/30HGO_A_ (**f**) after cone calorimetry tests.

**Figure 11 materials-14-01184-f011:**
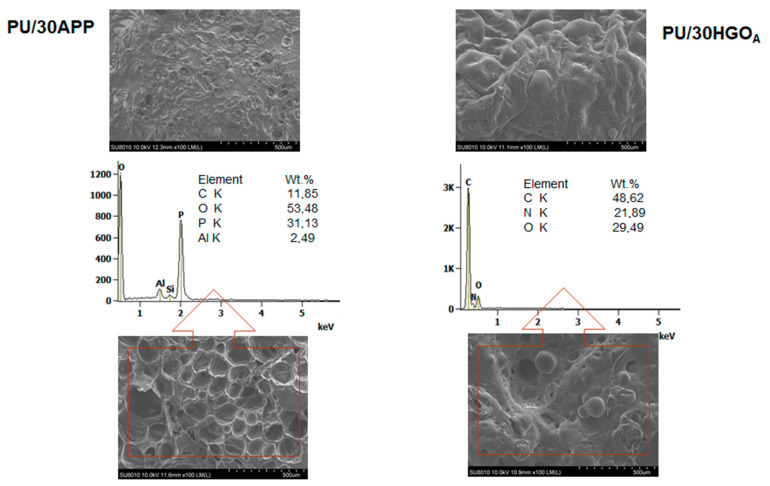
SEM images of PU/30APP and PU/30HGO_A_ after CC tests (outer and inner part of char) and EDS results.

**Figure 12 materials-14-01184-f012:**
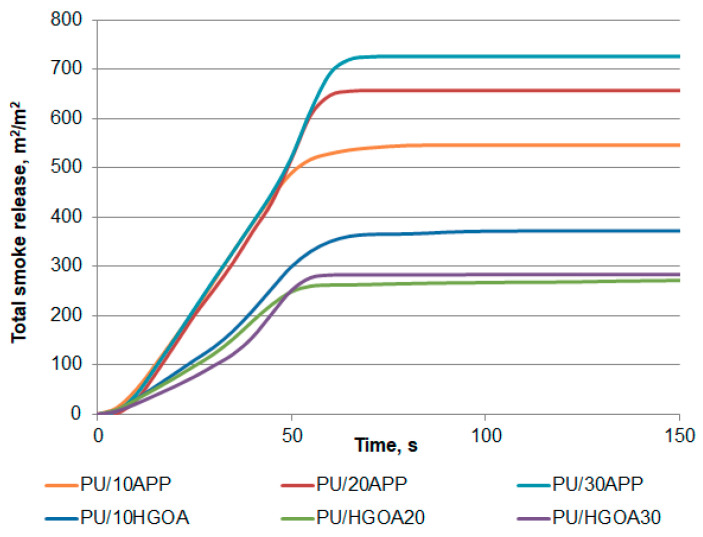
Representative curves of total smoke release of polyurethane foams with APP and HGO_A_ system.

**Figure 13 materials-14-01184-f013:**
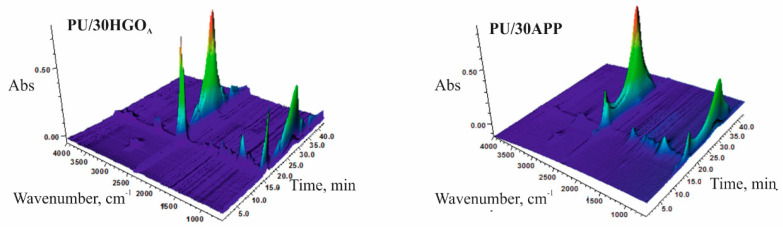
The 3D TGA/FT-IR spectra of PU/30HGO_A_ and PU/30APP.

**Figure 14 materials-14-01184-f014:**
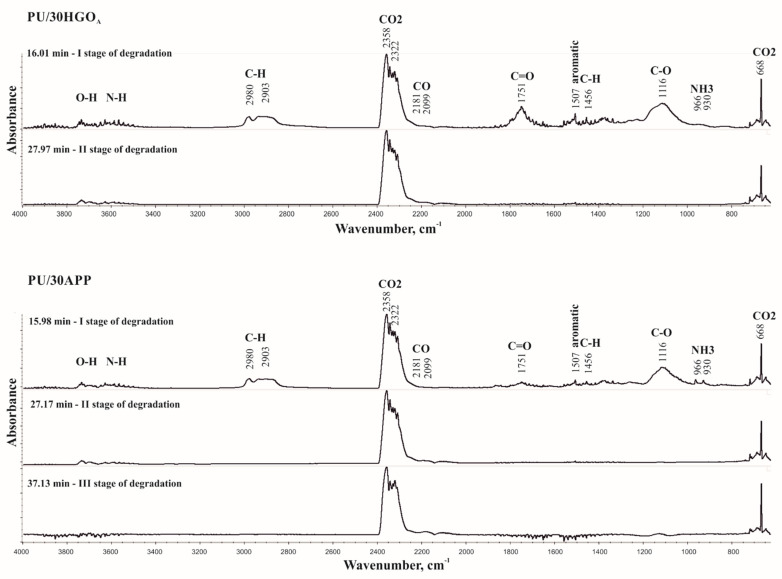
The 3D TGA/FT-IR spectra of PU/30HGO_A_ and PU/30APP.

**Figure 15 materials-14-01184-f015:**
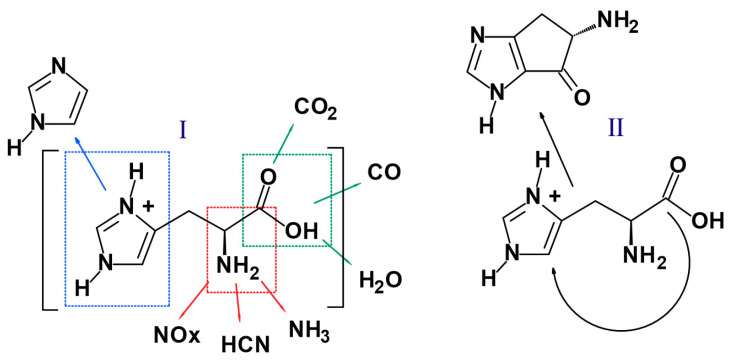
Proposed decomposition mechanism for histidine.

**Table 1 materials-14-01184-t001:** Formulations of flame retarded PU.

Samples	Petol 400, g	Petol 480, g	PC CAT TD-33, g	PC41, g	Niax Silicone L-6900, g	Ongronat 2100, g	APP, g	H, g	GO_A_, g
PU/10APP	80	20	1	0.5	2	110	10	0	0
PU/20APP	80	20	1	0.5	2	110	20	0	0
PU/30APP	80	20	1	0.5	2	110	30	0	0
PU/10HGO_A_	80	20	1	0.5	2	110	0	9	1
PU/20HGO_A_	80	20	1	0.5	2	110	0	19	1
PU/30HGO_A_	80	20	1	0.5	2	110	0	29	1

**Table 2 materials-14-01184-t002:** Process times for PU/APP and PU/HGO_A_ foams.

Samples	Process Parameter		
	Start Time, s	Gel Time, s	Tack-Free Time, s
PU/10APP	43	141	181
PU/20APP	32	135	182
PU/30APP	34	135	197
PU/10HGO_A_	22	124	203
PU/20HGO_A_	19	165	189
PU/30HGO_A_	21	164	193

**Table 3 materials-14-01184-t003:** The thermo-oxidative properties of H and GO_A_.

Component	Water Content, %	T_5%_, °C	T_50%_, °C	Residue at 950 °C, %
H	0.1	262	540	0
GO_A_	12.5	57	437	16

**Table 4 materials-14-01184-t004:** Maximum values of absorption signals for PU/APP and PU/HGO_A_ foams.

PU/ 10APP	PU/ 20APP	PU/ 30APP	PU/ 10HGO_A_	PU/ 20HGO_A_	PU/ 30HGO_A_	Bond Type (Vibration)
3304	3304	3304	3303	3303	304	N-H (stretching)
2978, 2925	2972, 2925	2975, 2927	2972, 2927	2972, 2927	2974, 2924	C-H (asymmetric stretching)
2870	2870	2870	2871	2871	2871	C-H (symmetric stretching)
2275	2277	2279	2277	2277	2277	N=C=O (stretching)
1708	1708	1707	1719, 1709	1720, 1709	1720, 1709	C=O (stretching)
1595	1595	1595	1595	1595	1595	Ar-H (deformation)
1518	1517	1517	1514	1514	1514	N-H (bending)
1455	1453	1455	1452	1451	1452	C-H (deformation)
1411	1411	1411	1411	1411	1411	PIR (deformation)
1219	1220	1220	1220	1220	1220	C-N (stretching)
1070	1071	1070	1071	1071	1071	C-O (stretching)
600–400	600–400	600–400	623	623	623	C-H (bending)

**Table 5 materials-14-01184-t005:** Physico-mechanical properties and main characteristics of the cellular structure of PU foams.

Samples	D, µm	Ar	Apparent Density, kg/m^3^	Friability, %
PU/10APP	513	1.4	40.7 ± 2.7	6.26
PU/20APP	558	1.2	41.5 ± 2.7	10.1
PU/30APP	534	1.2	44.9 ± 1.7	12.23
PU/10HGO_A_	567	1.5	33.1 ± 2.8	12.23
PU/20HGO_A_	666	1.4	30.2 ± 1.7	26.25
PU/30HGO_A_	616	1.4	36.4 ± 2.0	30.38

**Table 6 materials-14-01184-t006:** TG and DTG data of polyurethane foams with APP or HGO_A_ system.

Samples	T_5%_, °C	T_50%_, °C	DTG1, °C; %/°C	DTG2, °C; %/°C	DTG3, °C; %/°C	Residue in 950 °C, %
PU/10APP	239	314	288; 12.42	517; 3.83	776; 1.11	1.7
PU/20APP	240	327	284; 12.74	513; 3.49	778; 1.42	0.8
PU/30APP	256	396	293; 13.6	524; 2.93	789; 1.89	0.9
PU/10HGO_A_	232	313	292; 11.89	518; 3.85	-	1.4
PU/20HGO_A_	221	320	289; 9.19	518; 3.82	-	2.7
PU/30HGO_A_	220	328	289; 8.69	520; 3.84	-	0.9

**Table 7 materials-14-01184-t007:** Cone calorimeter results of polyurethane foams with APP or HGO_A_ system.

Samples	TTI, s	pHRR, kW/m^2^	MARHE, kW/m^2^	THR, MJ/m^2^	Residue, %
PU/10APP	6 (1 ^a^)	393 (4)	292 (20)	19 (0)	19.1 (8)
PU/20APP	6 (0)	419 (25)	308 (23)	19 (3)	19.6 (1)
PU/30APP	6 (1)	408 (20)	309 (13)	19 (2)	19.4 (4)
PU/10HGO_A_	5 (1)	324 (15)	249 (12)	19 (1)	23.0 (5)
PU/20HGO_A_	6 (1)	338 (22)	255 (8)	16 (1)	28.3 (10)
PU/30HGO_A_	6 (0)	336 (13)	260 (7)	17 (2)	19.0 (5)

^a^ The values in parentheses are the standard deviations.

**Table 8 materials-14-01184-t008:** Smoke emission of unmodified PU and polyurethane foams with APP or HGO_A_ system.

Samples	TSR, m^2^/m^2^	SEA, m^2^/kg	Ds_max_	VOF4
PU/10APP	586 (72)	622 (60 ^a^)	142 (10)	308 (15)
PU/20APP	685 (134)	739 (16)	158 (29)	369 (82)
PU/30APP	706 (118)	757 (52)	180 (7)	431 (4)
PU/10HGO_A_	388 (18)	466 (12)	108 (6)	233 (5)
PU/20HGO_A_	298 (13)	359 (79)	127 (18)	226 (2)
PU/30HGO_A_	295 (39)	381 (27)	176 (24)	222 (6)

^a^ The values in parentheses are the standard deviations.

## Data Availability

The data presented in this study are available on request from the corresponding author.

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
