# Peer review of "Burning Behaviour of Rigid Polyurethane Foams with Histidine and Modified Graphene Oxide"

_materials, 2021, doi:10.3390/ma14051184_

Round 1

Reviewer 1 Report

The manuscript is conceptualized and written very well. However, there are few minor issues that need to be addressed.

Page 5, line 212: the “-1” is missing on “cm” (spectral range).

On the Fig 3 (TG) and Fig 4 (FT-IR) authors showed all samples except the neat PU. Why not? The same valid for the Fig. 5.

On the Fig 4 the sample names are incorrect, need to be checked and corrected.

On the Fig. 6 the corresponding DTG curves are not showed, but only TG ones. What is showed on the 6(a)?

In Table 6, TG and DTG data of unmodified PU and polyurethane foams with APP or HGOA system are showed. However, the residual mass after 950 ºC is a little bit confusing: PU/10HGO A 1.4%, PU/20HGO A 2.7% and PU/30HGO A 0.9%. Is this the typo mistake? If not, why didn’t authors commented this?

Once again, in order to visually compare obtained results it is advised to show the corresponding results for the neat PU on the Figs. 6, 8, 9, 10, 12 and 14. The same valid for all tables in the manuscript.

Page 14, lines 364 and 369: the number of Fig is not 5 but 9.

Author Response

We would like to thank the Reviewer for all the valuable remarks. We agree with the recommendations that the manuscript should be improved and so the effort has been made to correct the article according to the comments.

  1. Page 5, line 212: the “-1” is missing on “cm” (spectral range).

The authors wish to thank the Reviewer for drawing attention to the error that was made.

  1. On the Fig 3 (TG) and Fig 4 (FT-IR) authors showed all samples except the neat PU. Why not? The same valid for the Fig. 5.

We appreciate the Reviewer’s suggestion. The aim of the work was to develop a new additive with better or at least similar properties to the best markets solutions. APP is one of the most influential and popular non-halogen fire retardants. For this reason, in te first stage of research the basic recipe has been modified in such a way to, allowing for the production of good quality material with a 30 wt. % of powder flame retardant system. The improvement of properties compared to pure PUR without flame retardants would not bring anything new in a field of fire retardancy PU foams. We are really sorry, but the test series described in the article was not conducted for pure foam (without the addition of flame retardants). 

  1. On the Fig 4 the sample names are incorrect, need to be checked and corrected.

We thank the Reviewer for the remarks. The suitable corrections in the figure according to the comment have been introduced.

  1. On the Fig. 6 the corresponding DTG curves are not showed, but only TG ones. What is showed on the 6(a)?

We thank the Reviewer for pointing that out. The Figures presenting graphs has been checked and corrected. In Figure 6a by mistake was TG curves from an analysis conducted in nitrogen. 

  1. In Table 6, TG and DTG data of unmodified PU and polyurethane foams with APP or HGOA system are showed. However, the residual mass after 950 ºC is a little bit confusing: PU/10HGO A 1.4%, PU/20HGO A 2.7% and PU/30HGO A 0.9%. Is this the typo mistake? If not, why didn’t authors commented this?

Thank the Reviewer for valuable comment. We carefully improved that inaccuracy in the manuscript. The description has been supplemented, and the degradation rate in individual stages has been referred to. Unfortunately, since the test was conducted in the air and the samples' mass was small, no linear dependence of the amount of residues on the proportion of flame retardants was observed. Such relationships were observed in the case of analyses conducted in nitrogen (please see the table below), which are not included in the article to not overextend it.

Samples

Residue in 950 °C, %

PU/10APP

13.1

PU/20APP

14.9

PU/30APP

15.2

PU/10HGOA

13.6

PU/20HGOA

15.6

PU/30HGOA

18.0

  1. Once again, in order to visually compare obtained results it is advised to show the corresponding results for the neat PU on the Figs. 6, 8, 9, 10, 12 and 14. The same valid for all tables in the manuscript.

We would like to thank the Reviewer for all his valuable remark. As reference materials for the developed flame retardant system, foams with the same amount (10, 20 and 30 wt%)  of one of the most popular non-halogen flame retardants used in polyurethane production were applied. The use of more samples as reference materials, in the author's opinion, would extend the article without adding new information. We are really sorry, but at the present stage of the workable, it is impossible to produce and test materials without FR additives under conditions comparable to those previously used and introduce them into the manuscript within the time allowed for reviews.

  1. Page 14, lines 364 and 369: the number of Fig is not 5 but 9.

Thank the Reviewer for valuable comment. We carefully improved that inaccuracy in the manuscript.

We have also corrected quite a number of other minor errors which we noticed while working on the text, and we believe that our manuscript in the present form can be published in the Journal.

Yours faithfully,

Kamila Salasinska,

Milena Leszczyńska

Reviewer 2 Report

The authors have demonstrated capability to obtain rigid polyurethane (PU) foams from non-halogen fire retardants system, containing histidine and modified graphene oxide and well characterized. The comparison of the histidine and graphene oxide-based structure PU foamed on the burning behavior were compared with commercial FR foam. The work, if published, will likely lead to great interests from the readers in this and other relevant fields. Some questions/comments for the authors to consider:

Can the authors comment on, would the rapid rotation in material structures affect the foaming process?

Overall, the reviewer would expect that more quantitative discussions/analysis regarding the fundamental mechanisms on the foaming could be conducted, given the rich information and unprecedented time resolution provided by the method. Perhaps this can be done in follow-up studies/analysis on these datasets to provide a more in-depth FR, beyond morphological description. Some of the potential future work on this may be discussed.

Author Response

Dear Reviewer,

We highly appreciate all the comments and find them very useful. We agree with the recommendations that the manuscript should be improved, so efforts have been made to correct the article according to the comments.

  1. Can the authors comment on, would the rapid rotation in material structures affect the foaming process? 

We thank the Reviewer for the comment and agree that the dispersion process has a significant impact on the course of the foaming process and the quality of the obtained materials' cell structure.

Filler particles tend to agglomerate, forming large aggregates in the PU matrix. In turn, too large particles can disturb the foaming process leading to the formation of materials with a heterogeneous structure. Moreover, large filler agglomerates can lead to gravitational drainage of cells in the initial foaming stage. The fillers' SEM images show a high degree of agglomeration of both H and GOA, thus to obtain a high degree of dispersion it was necessary to prepare a polyol-solid filler mixture using a mechanical stirrer and an ultrasonic disintegrator; however, the mixing time was intentionally reduced to a minimum. Moreover, snce an uncontrolled increase in the temperature of the mixed material, which may lead to changes in the chemical structure of the components, thus leading to difficulties at the stage of the foaming process and changes in the properties of the composite, the temperature of the mixture was controlled and cooled in an ice-water bath when the temperature was exceeded 50 °C.

Thanking the Reviewer for calling our attention to the issue the authors would like to say that every effort has been made to improve the description of the results.

  1. Overall, the reviewer would expect that more quantitative discussions/analysis regarding the fundamental mechanisms on the foaming could be conducted, given the rich information and unprecedented time resolution provided by the method. Perhaps this can be done in follow-up studies/analysis on these datasets to provide a more in-depth FR, beyond morphological description. Some of the potential future work on this may be discussed.

We thank the Reviewer for the valuable comment. This report focused on the application potential of the developed polyurethane composite materials regarding their fire retarding properties. Broader studies concerning the more fundamental aspects of foam growth processes are currently underway and will be a subject of further manuscripts, as suggested by the Reviewer. However, to authenticate the process carried out, while maintaining the article's consistency and brevity, a table with the process times has been added to the part describing the production of foams.

We have also corrected quite a number of other minor errors which we noticed while working on the text, and we believe that our manuscript in the present form can be published in the Journal.

Yours faithfully,

Kamila Salasinska,

Milena Leszczyńska
